# A Novel Mis-Seg-Focus Loss Function Based on a Two-Stage nnU-Net Framework for Accurate Brain Tissue Segmentation

**DOI:** 10.3390/bioengineering11050427

**Published:** 2024-04-26

**Authors:** Keyi He, Bo Peng, Weibo Yu, Yan Liu, Surui Liu, Jian Cheng, Yakang Dai

**Affiliations:** 1Suzhou Institute of Biomedical Engineering and Technology, Chinese Academy of Sciences, Suzhou 215163, China; hky2202104051@163.com (K.H.); pengbo@sibet.ac.cn (B.P.); liuyan@sibet.ac.cn (Y.L.); liusurui@sibet.ac.cn (S.L.); 2The School of Electrical and Electronic Engineering, Changchun University of Technology, Changchun 130012, China; yuweibo@ccut.edu.cn; 3State Key Laboratory of Complex & Critical Software Environment, Beihang University, Beijing 100191, China; 4International Innovation Institute, Beihang University, 166 Shuanghongqiao Street, Pingyao Town, Yuhang District, Hangzhou 311115, China

**Keywords:** deep learning, brain tissue segmentation, nnU-Net, loss function

## Abstract

Brain tissue segmentation plays a critical role in the diagnosis, treatment, and study of brain diseases. Accurately identifying these boundaries is essential for improving segmentation accuracy. However, distinguishing boundaries between different brain tissues can be challenging, as they often overlap. Existing deep learning methods primarily calculate the overall segmentation results without adequately addressing local regions, leading to error propagation and mis-segmentation along boundaries. In this study, we propose a novel mis-segmentation-focused loss function based on a two-stage nnU-Net framework. Our approach aims to enhance the model’s ability to handle ambiguous boundaries and overlapping anatomical structures, thereby achieving more accurate brain tissue segmentation results. Specifically, the first stage targets the identification of mis-segmentation regions using a global loss function, while the second stage involves defining a mis-segmentation loss function to adaptively adjust the model, thus improving its capability to handle ambiguous boundaries and overlapping anatomical structures. Experimental evaluations on two datasets demonstrate that our proposed method outperforms existing approaches both quantitatively and qualitatively.

## 1. Introduction

Brain tissue segmentation is a pivotal medical image processing technique employed to delineate and distinguish various tissues within the brain, including gray matter, white matter, and cerebrospinal fluid [1]. Its significance lies in its contribution to the diagnosis, treatment, and investigation of brain disorders, facilitating more precise identification of brain lesions and the formulation of effective treatment strategies [2]. Thus, precise brain tissue segmentation holds paramount importance in both medical research and clinical practice, as it not only deepens our comprehension of brain disorders but also fosters the refinement of diagnostic approaches and therapeutic interventions.

Current deep learning methods in medical image segmentation primarily use loss functions to measure the disparity between network predictions and Ground-Truth labels, guiding parameter updates [3]. However, these loss functions typically concentrate on segmentation in the whole image, calculating loss in a global region, which is known as global loss. The application of global loss often leads to the neglect of certain local regions in the image, including boundaries or mis-segmentation regions. Existing deep learning methods prioritize global region feature extraction and overall segmentation optimization, neglecting local region information at boundaries in magnetic resonance imaging (MRI), causing segmentation results to appear blurry or discontinuous. The complexity of segmentation tasks in brain tissue segmentation is due to the adjacency of different tissues and varying intensity characteristics, posing challenges for traditional loss functions. Consequently, segmentation errors such as mis-segmentation, omission, or over-segmentation at boundaries compromise overall segmentation accuracy. In short, despite significant progress, their focus is on the global region, neglecting local regions where segmentation errors often occur.

The segmentation of brain tissue relies heavily on local regions due to overlapping boundaries between different brain tissues. The local region is crucial for accurate brain tissue segmentation tasks. Gray matter, white matter, and cerebrospinal fluid often exhibit overlapping intensities and textures in MRI images, complicating the distinction between them. The boundaries between gray and white matter are typically indistinct, as they share similar intensity and texture characteristics. Currently, some methods calculate loss based on local regions (known as local loss), often focusing on boundary regions for loss calculation. Although boundary regions are frequently prone to errors in brain tissue segmentation, the regions where mis-segmentations occur are not confined to these edges alone. Therefore, it is essential to conduct research on loss calculation methods specifically targeting mis-segmentation.

The aim of this study is to address the limitations of existing deep learning methods in brain tissue segmentation, particularly regarding the oversight of crucial information at image boundaries. Recognizing the significance of accurate boundary delineation, especially in MRI images for diagnosing brain tissue abnormalities, our research endeavors to design a novel loss function that effectively leverages both global and local regional information based on an improved nnU-Net baseline architecture. The two-stage training strategy serves to address segmentation errors effectively. In the first stage, the network is optimized by calculating the loss of the global region, while the second stage involves defining a mis-segmentation loss function to adaptively adjust the model, thus improving its capability to handle ambiguous boundaries and enhancing segmentation accuracy.

The main contributions are as follows:We propose a novel mis-seg-focus (MSF) loss function that guides the network to focus on mis-segmented regions, improving the segmentation accuracy of mis-segmented regions. This allows for more precise segmentation of brain tissues.We propose a two-stage training strategy. In the first stage, the network undergoes training using global loss to grasp overall structural information. In the second stage, we use the MSF loss, enabling the network to train based on both global and local information. The global loss assists the network in capturing global information, while the MSF loss focuses on refining the mis-segmentation regions.We validate the efficiency of the proposed method on different datasets, including a dHCP dataset about infants and an OASIS dataset about the elderly population.

## 2. Related Work

Traditional brain tissue segmentation methods mostly rely on image processing techniques and machine learning algorithms, such as thresholding, region growth, and graph cuts. These methods achieve segmentation through manually designed features and rules. Tools like FSL and FreeSurfer [4,5], for instance, employ such methods to process adult brain tissue images. Additionally, there are tools specifically designed for pediatric brain tissue segmentation, such as iBEAT [6]. However, the effectiveness of these traditional methods is often influenced by factors like image quality and noise, and they exhibit certain limitations when dealing with complex scenarios involving blurry boundaries and overlapping anatomical structures.

With the rapid development of deep learning, significant progress has been made in brain tissue segmentation methods based on deep learning [1]. These methods use deep learning models such as Convolutional Neural Networks (CNNs) to learn feature representations from large amounts of data, enabling them to automatically learn more discriminative features and achieve better performance in brain tissue segmentation tasks [7,8,9]. In order to address the challenges encountered in brain tissue segmentation, researchers are continuously exploring new methods. Currently, the application of deep learning in brain tissue segmentation primarily relies on U-Net and its derivative networks [10,11,12,13], such as dual-encoder U-Net [14], dynamic U-Net [15], and U-Net integrated with fuzzy information [16]. Additionally, there are brain tissue segmentation networks based on improved Transformer structures, including Swin UNETR [17], which can extract features at different resolutions, and Transformer structures with varying weight configurations [18]. Fabian Isensee and his colleagues have developed a self-configuring method called nnU-Net based on U-Net [19]. On brain tissue segmentation datasets, the excellent architecture of nnU-Net enables it to achieve outstanding performance, demonstrating its strong adaptability and efficiency [20]. Despite the remarkable achievements of deep learning in the field of brain tissue segmentation, it still faces some challenges. Current deep learning methods tend to overlook mis-segmentation regions when processing segmentation results, which leads to suboptimal performance in handling complex scenarios such as blurry boundaries and overlapping anatomical structures.

In current brain tissue segmentation tasks, adjustments to the loss function have been shown to effectively improve the segmentation performance of the network. For instance, Zhang, X. and his team have introduced contrastive learning loss, using feature maps as key inputs for contrastive learning [21]. Meanwhile, Wan, X. and others use preset total loss functions to update the parameters of the student model [22]. Additionally, calculating loss based on edge regions is also a commonly used approach [23,24]. However, these methods still need improvement in guiding the network to more precisely focus on mis-segmentation regions at the edges of multi-label brain tissue.

Two-stage training methods have demonstrated their strengths in image segmentation, particularly in refining edge regions and enhancing segmentation accuracy. Typically, these methods involve a first stage of rough segmentation followed by a second stage of fine segmentation, making them suitable for challenging tasks such as vessel segmentation [25]. Additionally, nnU-Net incorporates a two-stage cascaded U-Net architecture that can further refine the coarse segmentation results obtained in the first stage. However, the current two-stage segmentation methods still need improvement in their ability to precisely segment mis-segmentation regions.

## 3. Materials and Methods

### 3.1. Overall Architecture

The architecture of the proposed framework is shown in Figure 1. The proposed model is based on nnU-Net. We improve the model structure by employing deeper filters to enhance feature extraction capabilities. We have proposed a novel mis-seg-focus (MSF) loss function, implemented by using a two-stage training strategy. In the first stage, we train the model using a global loss to locate mis-segmentated regions. In the second stage, we employ the mis-segmentation region extraction block (MS block). This block, at the end of each training epoch, calculates the intersection over Union between the network predictions and the corresponding Ground-Truth labels, thereby precisely extracting mis-segmentation regions for each label. Subsequently, we perform dilation on these regions and merge the mis-segmentation regions from different brain tissues. Finally, we compute the loss on these processed mis-segmentation regions, incorporating both global loss and our proposed MSF loss for the second stage of training.

### 3.2. Improved Network Architecture

The nnU-Net framework is an improved version based on the U-Net architecture, featuring a U-shaped design consisting of an encoder and a decoder. We have made further adjustments to the network architecture based on the original nnU-Net framework, deepening the overall network structure and increasing the number of filters in the encoder. In addition, we have adopted an asymmetric network design to better adapt to multi-label datasets, thereby enhancing the network’s capacity and expressive ability. The network incorporates a five-round downsampling dual convolution process. Notably, during the first three rounds of downsampling, the depth of the filter layers gradually doubles, significantly increasing the total number of network parameters. Furthermore, skip connections are utilized to fuse feature information from different network levels, improving feature extraction efficiency. Simultaneously, the deep supervision module collects feature maps of different sizes generated during the downsampling process and integrates this information into the loss function calculation.

### 3.3. Mis-Seg-Focus (MSF) Loss

#### 3.3.1. Mis-Segmentation Region Extraction

The extraction of mis-segmentation regions in the MSF loss is accomplished by the MS region extraction module. There are some specific regions formed by mis-segmentation between the network result and the Ground Truth. In order to more accurately identify and extract these regions, we utilize the rich pixel information contained in the last three stages of the upsampling process. We extract feature maps from these stages and apply the sigmoid function to compress the values between 0 and 1, making it easier to represent probabilities or perform binarization. Finally, the processed feature maps are output to the MS region extraction module to locate and extract the mis-segmentation regions in the MSF loss.

To facilitate region calculation and the extraction of mis-segmentation, the feature layer is first processed using argmax to convert it into three-dimensional data $y_{n}^{’}$. As shown in the global block diagram in Figure 1, the $y_{n}$ data are separated by labels, where N represents the number of labels, resulting in N label output results from $y_{1}^{’}$ to $y_{N}^{’}$. The same operation is performed on the Ground Truth (GT) to obtain GT $y_{1}$ to $y_{N}$ for N labels. Label extraction is performed separately for the N labels, yielding the mis-segmented region $M_{n}$ for the nth label, as shown in Equation (1).
(1)$$M_{n}=(y_{n}\cup {\hat{y}}_{n}^{\prime })-(y_{n}\cap {\hat{y}}_{n}^{\prime })$$

Due to the need for partial global regions during region calculation, the mis-segmentation regions of each label are dilated to obtain $M_{dn}$. The mis-segmentation regions of all labels are superimposed to obtain the global mis-segmentation region, as shown in Equation (2).
(2)$$M_{d}=\sum_{n=1}^{N}M_{dn}$$

#### 3.3.2. Mis-Seg-Focus (MSF) Loss Function

Mis-seg-focus (MSF) loss is a method of calculating loss specifically targeted at mis-segmentation regions. By utilizing this approach, the network can more effectively concentrate on those areas that have been incorrectly segmented, thereby enhancing the accuracy of segmentation.

Set A as the set of all pixel points in the whole MRI image, and extract $M_{d}$ as the set of all discrete segmentation error regions from the MS region, which is a subset of A, $M_{d}\subset A$. Therefore, the network output and GT for the mis-segmentation region are $\hat{y}’$ and $\hat{y}’$, as shown in Equations (3) and (4). The $y^{\prime }$ and y regions in the formula belong to the global region set A, while the $\hat{y}’$ and $\hat{y}$ regions belong to the mis-segmentation region set $M_{d}$.
(3)$$\hat{y}=M_{d}⊙y$$
(4)$$\hat{y}\prime =M_{d}⊙y^{\prime }$$

The segmentation loss function uses a compound loss consisting of Dice loss and Cross Entropy loss. The segmentation loss function is shown in Equation (5). The global region loss calculation is shown in Equation (6), and the MSF loss calculation is shown in Equation (7).
(5)$$L_{loss}(y_{1},y_{2})=\sum_{j=1}^{dn}\alpha_{j}[L_{Dice}(y_{1},y_{2})+\omega *L_{CE}(y_{1},y_{2})]$$
(6)$$L_{Global}=L_{loss}(y,y^{\prime })$$
(7)$$L_{MSF}=L_{loss}(\hat{y},\hat{y}\prime )$$

As shown in Equation (5), $y_{1}$ and $y_{2}$ in the loss function $L_{loss}$ represent the Ground Truth and the network’s prediction results in the region, respectively, where weights ω are used to balance the three loss functions. The Dice loss function is shown in Equation (8), and the CE loss function is shown in Equation (9).
(8)$$L_{Dice}(y,y^{\prime })=-\frac{2\sum_{i=1}^{N}(y_{i}*y_{i}^{\prime })}{\sum_{i=1}^{N}y_{i}+\sum_{i=1}^{N}y_{i}^{\prime }}$$
(9)$$L_{CE}(y,y^{\prime })=-\frac{1}{N}\sum_{i=1}^{N}\left[y_{i}\log(y_{i}^{\prime })+(1-y_{i})\log(1-y_{i}^{\prime })\right]$$
where $a_{j}$ in Equation (5) represents the weight of the deep supervision module, and the specific formula is shown in Equation (10), where dn represents the total number of layers in the deep supervision module, and j=1 represents the first layer in the decoding stage. The feature maps in the decoding stage are all output through Sigmoid and added to the loss function, and their weights increase as the size of the upsampling feature map increases, allowing gradient information to be injected deeper into the network.
(10)aj=1[∑j=1dn12(j−1)]2(j−1)

### 3.4. Two-Stage Training Strategy

By implementing a two-stage training strategy, we can effectively handle difficult regions in segmentation.

In the 3D cascade network in nnU-Net, although the first U-Net operates on downsampling images and the second U-Net can be trained to refine the segmentation map created at full resolution by the previous one, it still only performs segmentation on the global target of the previous stage, without significant improvement for difficult segmentation of tissue edges. To address this issue, we adopted a two-stage segmentation network joint training strategy.

In the first stage, the network undergoes global training to initially determine the segmentation results, using a global loss to perform coarse segmentation on images, as shown in Equation (11). When evaluation metrics such as Dice and HD95 cannot be improved, the training set has been fully fitted. At this point, there are some difficult-to-distinguish regions in the mis-segmentation and global areas. Through matrix calculation of the network output results and Ground Truth, the mis-segmentation regions are formed. In this study, the first-stage training epoch is 300.

In the second stage, an MSF loss function is used in the network to constrain and correct the erroneous segmentation regions that appeared in the first stage. A novel MSF loss is proposed, as shown in Equation (12). The proportion of global loss and MSF loss is weighted through the weight λ. This staged approach enables the network to more accurately handle complex segmentation tasks, especially in difficult regions of tissue boundaries.
(11)$$L_{loss1}=L_{Global}$$
(12)$$L_{loss2}=L_{Global}+\lambda *L_{MSF}$$

## 4. Results

### 4.1. Datasets

Two sets of data were used to evaluate the feasibility of the proposed method. One is from the Developing Human Connectome Project (dHCP), and the other is from the Open Access Series of Imaging Studies (OASIS). Informed consent was obtained from the imaging subjects in compliance with the Institutional Review Board.

The dHCP dataset is an international collaborative project aimed at studying infant brain development. All infants were born at term (37–44 weeks of gestation) and underwent imaging. It included T1 modal data from 40 neonatal subjects, axial slice stacks with an in-plane resolution of 0.8 × 0.8 mm^2^, and 1.6 mm slices overlapping by 0.8 mm. The Ground Truth contains nine labels, including gray matter, white matter, cerebrospinal fluid, background, ventricles, cerebellum, deep gray matter, brainstem, and hippocampus. The dataset is split into training and testing sets in a 4:1 ratio, and all training data underwent 5-fold cross-validation.

The OASIS dataset is a collection of MRI datasets used to study brain structure and function in the adult population. This group comprises 416 subjects aged between 18 and 96 years. Each subject has three or four separate T1-weighted MRI scans obtained during a single scanning session. All the subjects are right-handed and include both males and females. The image dimensions of the OASIS dataset are 208 × 176 × 176 (height × width × slices), where each axial scan encompasses 176 slices. The Ground Truth contains four labels, including gray matter, white matter, cerebrospinal fluid, andcortex. The dataset is split into training and testing sets in a 4:1 ratio, and all training data underwent 5-fold cross-validation.

### 4.2. Evaluation Metrics

We adopted the Dice coefficient (Dice) and the 95th percentile of the Hausdorff distance (HD95) as the primary evaluation metrics to comprehensively evaluate the performance of the proposed methods.

The Dice coefficient, as an effective indicator for measuring the similarity between two samples, is used to quantify the degree of overlap between the model prediction result and the true label, thereby accurately reflecting the segmentation accuracy of the model. This indicator pays particular attention to the consistency of the overall pixel distribution, that is, the degree of agreement between the model prediction result and the actual brain tissue structure.

The HD95 focuses on measuring the fineness of the model in boundary processing. It reflects the model’s ability to grasp boundary details by calculating the maximum mismatch between the model segmentation result and the true value, that is, the furthest distance between two sets of points. Since Hausdorff distance is extremely sensitive to boundary anomalies in segmentation results, HD95 can effectively reveal minor deviations in the boundary processing of the model.

### 4.3. Implementation Details

Our improved model is implemented in PyTorch 2.0, a GPU-based neural network framework, and optimized based on the nnU-Net integration framework. Code testing and network training were performed on Ubuntu 20.04 systems and accelerated using NVIDIA RTX3090Ti (with 24GB of video memory) GPUs. The following are the hyperparameter settings for the experiment.

The network training framework uses the 3D U-Net network framework in nnU-Net, and the optimizer uses Adam with a Leaky ReLU activation function. The initial learning rate is 3 × 10^−4^, and there are 250 batches per epoch with a batch size of two. The learning rate training strategy is to calculate the exponential moving average loss of the training set and the validation set. If the exponential moving average loss of the training set does not decrease by 5 × 10^−3^ within the last 30 epochs, the learning rate is attenuated by a factor of five.

### 4.4. Comparison Experiment

#### 4.4.1. Experiment Settings

We employed several state-of-the-art deep learning methods (including 3D U-Net, DynUNet, and Swin UNETR) that are based on the U-Net structure, all of which feature 3D network architectures. Additionally, we utilized two publicly available datasets: the dHCP dataset, containing nine labels (including gray matter, white matter, cerebrospinal fluid, background, ventricles, cerebellum, deep gray matter, brainstem, and hippocampus), and the OASIS dataset, containing four labels (including gray matter, white matter, cerebrospinal fluid, and cortex). To ensure the fairness of the experiments, all methods used the same compound loss function for network training, with 600 training epochs. Due to the difficulty of ensuring fairness in data augmentation and network parameters among various algorithms, we tried to use the original source code in the comparative experiments and adopted similar parameter settings.

#### 4.4.2. Segmentation Results on dHCP

As shown in Table 1, the performance of the original nnU-Net parameter framework is compared with the enhanced nnU-Net network parameter framework (referred to as DP, shorthand for “deep network”). The experimental results indicate that across various tissues, the deep network exhibits significant improvements in both the Dice coefficient and HD95 metrics. Furthermore, when MSF loss (MSF) is incorporated alongside DL, the experimental outcomes are further enhanced. Compared to the original nnU-Net, our proposed method achieves an improvement of 0.4 percentage points in the mean Dice metric and a reduction of 1.3 in the mean HD95 metric, with notable improvements across all tissues. Especially for small labeled tissues, like the hippocampus and ventricles, the segmentation performance is optimal. When compared to other state-of-the-art deep learning approaches and commonly used traditional methods (BET2), our method outperforms them by 5.8–8.8 percentage points in the mean Dice metric and achieves a reduction of 5.3–9.6 in the mean HD95. These findings convincingly demonstrate the superiority of our approach to brain tissue segmentation tasks.

As shown in the visualization results in Figure 2, we performed visualization on the coronal, sagittal, and axial planes and presented the multi-label segmentation results and mis-segmentation results (red areas) of the network. Due to nnU-Net’s excellent post-processing module, the edges of its segmentation results are smoother compared to other segmentation methods. The segmentation results obtained by our proposed method are very close to the Ground-Truth labels, with good edge segmentation effects, especially in the multi-label intersection areas. By comparing the red mis-segmentation areas, it can be clearly seen that our method produces the fewest mis-segmentation areas compared to other comparison methods.

#### 4.4.3. Segmentation Results on OASIS

To further validate the generalization capability of our two-stage nnU-Net training method based on MSF loss, we tested our model on the OASIS dataset. The OASIS brain tissue segmentation has a total of four segmentation labels, namely gray matter, white matter, cerebrospinal fluid, and cortex, with the last column representing the average of the four labels. As shown in Table 2, our method achieved the best segmentation performance in terms of average Dice and average HD95 compared to other segmentation networks. Compared to segmentation networks such as DynUNet and Swin UNTER, the Dice improved by approximately 2 to 3 percentage points, while the HD95 varied between approximately 0.2 and 3.63. Compared to the original nnU-Net framework, our method improved Dice and HD95 by 0.61 and 0.5032, respectively, and there was an improvement in the evaluation metrics for each label.

As shown in the visualization results in Figure 3, we have performed visualization on the coronal, sagittal, and axial planes and presented the multi-label segmentation results and mis-segmentation results (red areas) of the network. These results demonstrate the effectiveness of our proposed method on different multi-label brain tissue segmentation tasks, improving the network’s edge segmentation capability and enhancing the accuracy of brain tissue segmentation.

### 4.5. Ablation Study

#### 4.5.1. Design of MSF Loss Function

To verify the effectiveness of the MSF loss and determine the optimal hyperparameter settings for mis-segmentation regions, we conducted a series of ablation experiments on MSF region extraction. Under the same training strategy and network hyperparameters, we trained multiple network models. As shown in Table 3, by comparing the results of nnU-Net and the best results, we confirmed that the MSF loss can improve the performance of the model, as reflected in a 0.31 increase in the Dice index and a 0.36 increase in the HD95 index. In the process of exploring the hyperparameters for mis-segmentation regions, we tried three different settings for the expansion coefficient: an expansion coefficient of 2, an expansion coefficient of 5, and an expansion coefficient of 10. After detailed testing, we found that the setting of the expansion coefficient of 2 weighted performed the best on the evaluation metrics, especially on the HD95 index, achieving good parameter optimization.

#### 4.5.2. Segmentation Loss Selection

We conducted a series of ablation experiments covering Dice loss, CE loss, and joint loss functions in order to determine the optimal brain tissue segmentation loss. We maintained the same nnU-Net network architecture, removed the influence of network architecture and two-stage training strategies, and relied solely on different loss functions for network training. For each experiment, we applied different loss functions to calculate the Dice evaluation metrics in Table 4 and visually displayed the comparison results in Figure 4. The joint loss Dice and CE performed best in terms of Dice scores across all tissues. We experimented with four different values for the weight λ of the mis-segmentation loss function, including 0.5, 1, 10, and 100. As shown in Figure 4, the best results were achieved when the weight was set to 1.

#### 4.5.3. Effect of the Two-Stage Training Strategy

To verify the effectiveness of the two-stage training strategy, we conducted tests under the same network parameters and hyperparameters for MSF loss, with both the first and second stages trained for 300 epochs. As shown in Table 5, by comparing Experiment 1 and Experiment 2, we observed that the network had already achieved a state of training fit, indicating that the evaluation metrics could not be further improved. Next, we tested the scenarios of using only global loss and only MSF loss in the second stage separately, and the results showed that the segmentation performance in both cases was inferior to using both global and MSF losses. In Experiment 4, although both stages employed a combination of global and MSF losses for training, the large mis-segmentation regions in the first stage led to a high variation rate in loss values, which affected the final segmentation results. In summary, it can be concluded that using global loss in the first stage and adopting MSF loss functions in the second stage can achieve the best segmentation results.

As shown in Figure 5, the first column in the figure shows the mis-segmentation regions of the whole brain (represented in red), while the second to fourth columns present the segmentation results for three tissues: the hippocampus, ventricle, and white matter, with yellow indicating correctly segmented regions and red indicating mis-segmentation regions. In the first stage, we adopted the segmentation results from the 100th epoch and the last epoch, and it can be clearly seen that there are still many mis-segmentation regions represented in red in the nnU-Net training results, which constitute the MSF loss regions. In the second stage, we applied both global and MSF losses, and it can be observed that as the epochs increase, the originally difficult-to-converge mis-segmentation regions gradually decrease. This observation strongly validates the effectiveness of combining global and MSF losses, as well as the two-stage training strategy.

#### 4.5.4. Ablation Study on Network Architecture

To verify the feasibility of the improved network framework, we conducted ablation experiments. In this part of the experiment, we adopted the same training strategy and loss function. As shown in Table 6, our proposed method achieved better segmentation results compared to the original network framework, specifically with a 0.9 increase in the Dice coefficient, and corresponding improvements were observed in every tissue.

## 5. Discussion

The dHCP dataset is specifically designed for labeled data collection for infant brain tissues, while the OASIS dataset provides labeled data for adult brain tissues. Through comparative experimental validation, our proposed method has achieved optimal performance on both of these publicly available datasets, fully demonstrating its wide applicability. Since the adult brain has fully developed, with high contrast between tissues, our method performs well in segmenting adult brain tissues. However, segmenting infant brain tissues poses more challenges due to their ongoing development and relatively small differences among various tissues, especially when it comes to smaller tissue segments. Nonetheless, our method is still capable of effectively segmenting infant brain tissues.

The MSF loss function focuses the network’s attention on mis-segmentation regions by calculating losses specifically for those areas. We have chosen Dice and Cross Entropy losses to calculate the losses for both mis-segmentation and global regions. Combined with a two-stage training strategy, the first stage involves training with global loss, enabling the network to learn global information until the evaluation metrics can no longer be improved. In the second stage, both global loss and MSF loss are used for training, allowing the network to acquire both global and local information simultaneously. Here, “local information” refers specifically to the mis-segmentation regions. After the first stage, there may be some mis-segmentation regions along tissue edges. Since this stage employs commonly used global loss functions, the network tends to focus on global information. In the second stage, the network is trained using both global and MSF losses, enabling it to concentrate on the mis-segmentation regions and, thereby, improve segmentation accuracy.

However, due to significant size differences among various brain tissues, especially for smaller structures like the hippocampus, which require high segmentation accuracy, we can adopt different mis-segmentation region extraction methods based on the size of the brain tissue in subsequent work.

## 6. Conclusions

Experimental results demonstrate that our proposed two-stage nnU-Net framework training method with MSF loss achieves optimal segmentation outcomes in brain tissue segmentation tasks. By adopting a deeper and wider network architecture, the model captures more intricate details and features in multi-label brain tissue segmentation. Compared to other advanced deep learning methods, including the original nnU-Net, our approach proves its effectiveness on publicly available datasets such as dHCP and OASIS, with improvements across various evaluation metrics. This indicates that our method excels at handling complex and variable brain tissue structures, enhancing segmentation accuracy. Notably, it stands out when dealing with intricate and diverse brain anatomies. Future work can explore optimizing the loss function and enhancing the generalization capabilities of the network model to tackle more challenging brain tissue segmentation tasks.

## Figures and Tables

**Figure 1 bioengineering-11-00427-f001:**
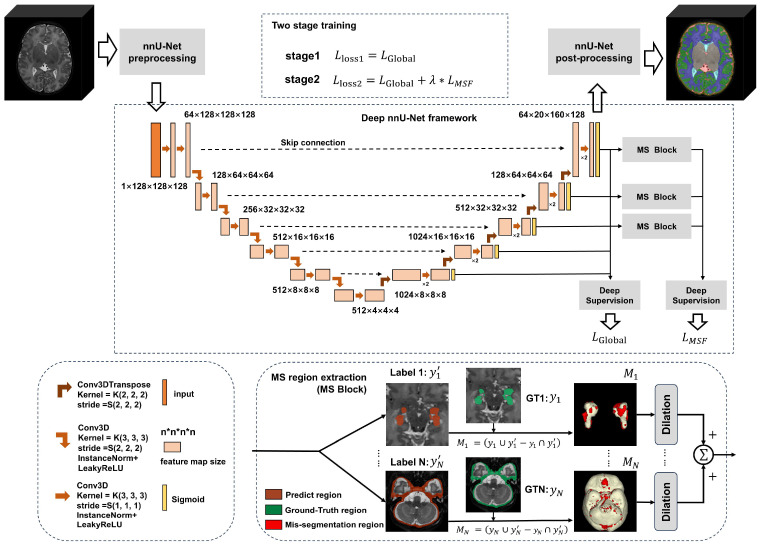
**Overall architecture of the multi-label brain tissue segmentation method.** It includes the enhanced nnU-Net main network framework (comprising nnU-Net network preprocessing, an enhanced nnU-Net network framework, and nnU-Net network post-processing), network framework information description, and MS region extraction section.

**Figure 2 bioengineering-11-00427-f002:**
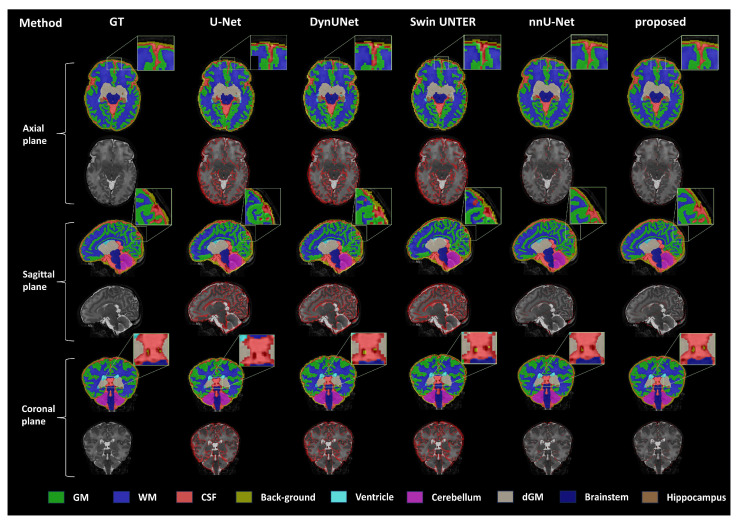
**Visualization of the segmentation results on the dHCP dataset**. Advanced deep learning brain tissue segmentation methods were compared with the nnU-Net baseline network and our proposed method. Each plane demonstrates the segmentation results of different methods, with the mis-segmentation regions (shown in red) displayed beneath the segmentation outcomes.

**Figure 3 bioengineering-11-00427-f003:**
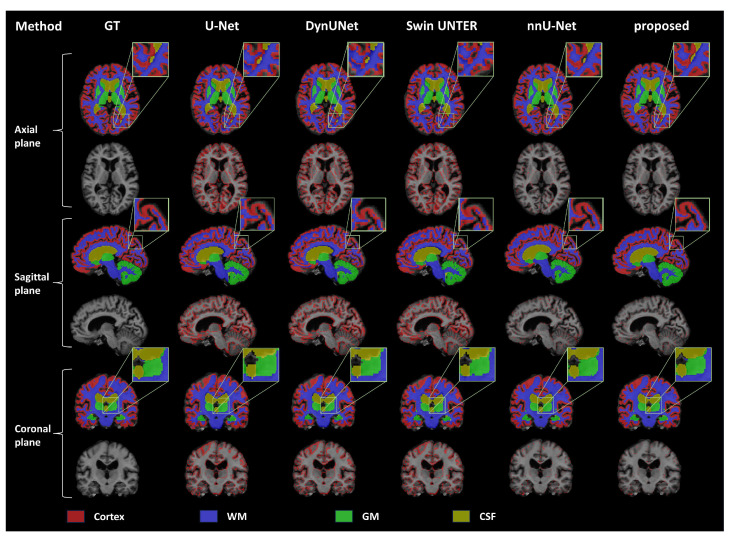
**Visualization of the segmentation results on the OASIS dataset**. Advanced deep learning brain tissue segmentation methods were compared with the nnU-Net baseline network and our proposed method. Each plane demonstrates the segmentation results of different methods, with the mis-segmentation regions (shown in red) displayed beneath the segmentation outcomes.

**Figure 4 bioengineering-11-00427-f004:**
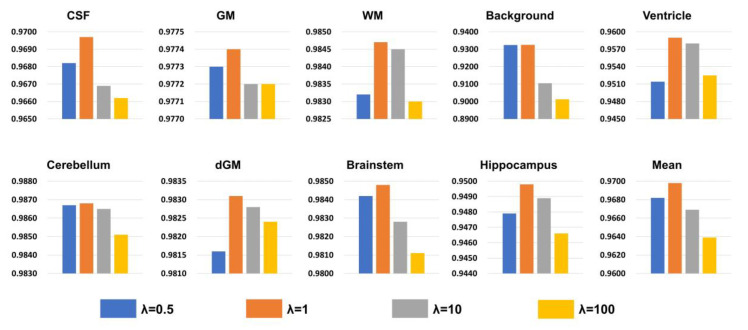
**Experimental results with different weights.** The ordinate of the coordinate chart is Dice as an evaluation metric, which shows a total of nine kinds of brain tissue and average results.

**Figure 5 bioengineering-11-00427-f005:**
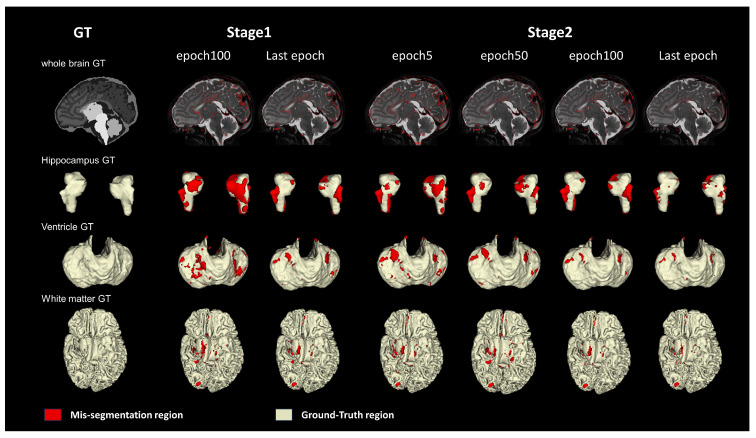
**Comparison of the mis-segmentation regions in different epochs of the two stages on the dHCP dataset.** Three types of brain tissues (hippocampus, ventricle, and white matter) and the whole brain are visualized. The last epoch is the final round result of this stage. The red region is the mis-segmentation region.

**Table 1 bioengineering-11-00427-t001:** **Quantitative evaluation results of the segmentation on the dHCP dataset.** The results of advanced deep learning methods and our proposed method with 5-fold cross-validation. Using Dice and HD95 as evaluation metrics, we show the segmentation results and mean results for nine brain tissues and present the best results in a deep representation.

Methods	Metrics	CSF	GM	WM	Background	Ventricle	Cerebellum	dGM	Brainstem	Hippocampus	Mean
U-Net	Dice	0.8711	0.8919	0.9103	0.8523	0.8518	0.9388	0.9524	0.9248	0.8227	0.8912
DynUNet		0.8957	0.9115	0.9389	0.8691	0.8752	0.9621	0.9567	0.9464	0.8878	0.9159
Swin UNETR		0.9077	0.9195	0.9433	0.8755	0.8861	0.9651	0.9528	0.9495	0.8620	0.9179
nnU-Net		0.9676	0.9752	0.9829	0.9314	0.9546	0.9841	0.9807	0.9816	0.9425	0.9667
proposed		0.9707	0.9782	0.9849	0.9341	0.9603	0.9879	0.9849	0.9848	0.9507	0.9707
U-Net	HD95	16.9027	21.8376	19.1212	24.1954	20.9448	16.8331	12.1901	10.9936	7.9166	16.7706
DynUNet		10.7693	18.6973	18.4298	20.5206	21.0848	14.4878	13.9697	8.4157	6.9788	14.8171
Swin UNETR		10.5709	16.3282	17.7961	20.3646	19.2571	17.5318	21.9199	20.3314	27.8359	19.1040
nnU-Net		15.8713	5.8862	6.9321	21.7541	14.1419	12.5969	10.9906	4.2461	4.4866	10.7673
proposed		15.1320	5.5934	6.7090	19.0621	7.5722	12.2246	11.0764	4.0602	3.8079	9.4709

**Table 2 bioengineering-11-00427-t002:** **Quantitative evaluation results of the segmentation on the OASIS dataset.** The results of advanced deep learning methods and our proposed method with 5-fold cross-validation. Using Dice and HD95 as evaluation metrics, we show the segmentation results and mean results for four brain tissues.

Methods	Metrics	Cortex	GM	WM	CSF	Mean
U-Net	Dice	0.9221	0.9471	0.9182	0.9269	0.9286
DynUNet		0.9342	0.9608	0.9654	0.9559	0.9541
Swin UNTER		0.9420	0.9524	0.9599	0.9488	0.9463
nnU-Net		0.9414	0.9653	0.9711	0.9603	0.9595
proposed		0.9497	0.9713	0.9782	0.9633	0.9656
U-Net	HD95	13.6718	11.2451	8.1022	9.8977	10.7292
Dyn U-Net		5.0010	5.8043	8.0469	9.9561	7.2021
Swin U-Net		11.2250	9.9171	6.7285	9.8577	9.4321
nnU-Net		5.6299	6.6751	6.7099	11.3947	7.6024
proposed		5.3959	6.4759	6.3539	10.1710	7.0992

**Table 3 bioengineering-11-00427-t003:** **Experiment on mis-segmentation region dilation factor on the dHCP dataset**. Dice and HD95 are used as evaluation metrics. dilation N: N represents the dilation factor.

	Metrics	CSF	GM	WM	Background	Ventricle	Cerebellum	dGM	Brainstem	Hippocampus	Mean
nnU-Net + dilation10	Dice	0.9681	0.9745	0.9834	0.9292	0.9557	0.9840	0.9811	0.9803	0.9448	0.9661
nnU-Net + dilation5		0.9695	0.9768	0.9843	0.9318	0.9585	0.9865	0.9822	0.9835	0.9494	0.9692
nnU-Net + dilation2		0.9697	0.9774	0.9847	0.9325	0.9590	0.9868	0.9831	0.9848	0.9498	0.9698
nnU-Net + dilation10	HD95	15.7291	7.1113	7.1927	19.2956	14.5516	15.2341	10.2597	4.8211	4.0154	10.9123
nnU-Net + dilation5		15.5127	12.9591	8.6446	21.3913	8.0140	12.1365	18.1506	3.9420	4.0858	11.6485
nnU-Net + dilation2		15.2935	6.2203	7.9551	19.7980	13.6532	12.4585	10.0997	3.9866	4.2801	10.4161

**Table 4 bioengineering-11-00427-t004:** **Experiment on the selection of the loss function on the dHCP dataset**. Dice is used as an evaluation metric.

Loss	CSF	GM	WM	Background	Ventricle	Cerebellum	dGM	Brainstem	Hippocampus	Mean
Dice	0.9645	0.9736	0.9818	0.9297	0.9502	0.9818	0.9767	0.9799	0.9331	0.9635
CE	0.9651	0.9713	0.9802	0.9319	0.9482	0.9852	0.9751	0.9801	0.9313	0.9632
Dice + CE	0.9676	0.9752	0.9829	0.9314	0.9546	0.9841	0.9807	0.9816	0.9425	0.9667

**Table 5 bioengineering-11-00427-t005:** **Results of the 5-fold cross-validation of the two-stage training ablation experiment on the dHCP dataset.** Dice is used as the evaluation metric. Global refers to the global loss, MSF refers to the MSF loss, and Total refers to the Global + MSF.

First Stage	Second Stage	CSF	GM	WM	Background	Ventricle	Cerebellum	dGM	Brainstem	Hippocampus	Mean
Global	NA	0.9676	0.9752	0.9829	0.9314	0.9546	0.9841	0.9807	0.9816	0.9425	0.9667
Global	Global	0.9657	0.9751	0.9838	0.9317	0.9546	0.9852	0.9803	0.9819	0.9427	0.9669
Global	MSF	0.9677	0.9764	0.9840	0.9105	0.9544	0.9854	0.9812	0.9806	0.9418	0.9647
Total	Total	0.9696	0.9772	0.9847	0.9289	0.9588	0.9866	0.9825	0.9840	0.9504	0.9692
Global	Total	0.9697	0.9774	0.9847	0.9325	0.9590	0.9868	0.9831	0.9848	0.9498	0.9698

**Table 6 bioengineering-11-00427-t006:** **Results of the 5-fold cross-validation of the network architecture improvement ablation experiment on the dHCP dataset.** Dice is used as the evaluation metric. The best results are highlighted.

	CSF	GM	WM	Background	Ventricle	Cerebellum	dGM	Brainstem	Hippocampus	Mean
nnU-Net	0.9697	0.9774	0.9847	0.9325	0.9590	0.9868	0.9831	0.9848	0.9498	0.9698
proposed	0.9707	0.9782	0.9849	0.9341	0.9603	0.9879	0.9849	0.9848	0.9507	0.9707

## Data Availability

dHCP: Developing Human Connectome Project (dHCP)|The Developing Human Connectome Project (developingconnectome.org), accessed on 4 July 2021,OASIS: OASIS Brains—Open Access Series of Imaging Studies (oasis-brains.org), accessed on 25 April 2007.

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
