# Peer review of "A Novel Mis-Seg-Focus Loss Function Based on a Two-Stage nnU-Net Framework for Accurate Brain Tissue Segmentation"

_bioengineering, 2024, doi:10.3390/bioengineering11050427_

Round 1
Reviewer 1 Report
Comments and Suggestions for Authors
This study introduces a novel loss function, MSF loss, to optimize segmentation errors in brain tissue segmentation. This study presents an interesting approach, however, several issues need to be addressed before final publication.
The description of the method overview, particularly how the mis-segmentation loss function is computed within the entire framework, lacks clarity and needs further elaboration.
The authors' innovation lies in proposing a new loss function, MSF loss, to optimize segmentation errors. It's crucial to elucidate the relationship between this loss function and local loss as well as global loss in detail.
The terms "local region", "global region", "critical region", and "boundary region" need to be clearly defined within the manuscript to ensure readers' understanding.
The definition of segmentation loss involves three loss functions: dice loss, ce loss, and iou loss. Why select these three losses and how their weights are set should be explicitly stated in the methodology section.
In Figures 2 and 3, each column represents different meanings such as GT, other methods, and proposed method. It's imperative to clarify what each row represents within the figures and legends. Additionally, the overlay of grayscale and red images beneath the labels requires explanation within the figures.
In Figure 4, it is recommended to present the combined results of the three loss functions in the last column for clarity.
Clear description is needed regarding whether the network's input is 2D or 3D images and whether the inputs of the comparative methods are 2D or 3D.
Given the utilization of two publicly available datasets, it's essential to compare the segmentation performance achieved by existing studies on these datasets. The results of this study should be juxtaposed with previous research to establish the efficacy of the proposed method.
The selection of datasets from different age groups, adults and infants, warrants justification. The performance of the proposed method in brain tissue segmentation tasks across different age groups, especially the prevalence of mis-segmented regions in infant brains, needs thorough explanation and discussion.
The division ratio of training, validation, and testing sets in the experiments needs specification, along with whether repeated experiments were conducted. Additionally, it's advisable to include statistical analysis results in the methodology to demonstrate the superior performance of the proposed method.
Comments on the Quality of English Language
The language of the manuscript requires improvement as many expressions are ambiguous and difficult to comprehend. Furthermore, the legends and headings in the figures and tables need to explicitly state which experiment's results they represent.
Reviewer 2 Report
Comments and Suggestions for Authors
This article discusses the significance of accurate brain tissue segmentation in the context of diagnosing, treating, and studying brain diseases. It highlights the challenge of distinguishing boundaries between different brain tissues due to overlapping regions, which can lead to errors in segmentation. The authors propose a novel approach based on a two-stage nnU-Net framework, focusing on addressing mis-segmentation along boundaries. In the first stage, a global loss function is utilized to identify mis-segmentation regions. In the second stage, a mis-segmentation-focused loss function is defined to adaptively adjust the model, enhancing its ability to handle ambiguous boundaries and overlapping anatomical structures. Experimental evaluations conducted on two datasets demonstrate that the proposed method outperforms existing approaches both quantitatively and qualitatively. This suggests that the approach is effective in improving the accuracy of brain tissue segmentation, particularly in challenging scenarios involving ambiguous boundaries and overlapping structures.
The article is well structured, the results are highly novel, a comparison is made with existing methods, and a good review is given that reflects the current state of the art.
The work can be considered for publication after addressing the following comments:
1. The discussion section needs to be expanded. It is necessary to reflect the limitations of the model and possible ways of developing the research. Provide a brief comparison with existing methods, highlighting the advantages and disadvantages of the proposed model.
2. It is necessary to provide a more detailed description of the evaluation metrics used, as well as the rationale for their use in this work. Make references to the work within which they were developed.
Round 2
Reviewer 1 Report
Comments and Suggestions for Authors
The paper can be accepted in present form.